**Comment**

# Participatory approaches should be used to address the ethics of social media experiments

Vincent J. Straub & Jason W. Burton

The use of social media field experiments has led to calls for revised ethical guidelines—but none have stuck. A participatory governance approach, similar to developments in AI, can improve practices and collectively align the interests of researchers, platforms and users.

Academic researchers make subjective, ethical judgments when designing experiments on human participants. This is particularly evident in the decisions psychologists must make when designing and implementing field experiments on social media. These experiments typically involve administering some treatment to users and analysing their online behaviour before and after the treatment[1]. They can be run with platform cooperation (e.g., with direct industry-academia collaboration, as in ref. [2]) or independently, without explicit collaboration agreements with the platform on which they are run (e.g., by having consenting users download web browser extensions that scrape posts from their feed, as in ref. [3]). They can also be run with participants' informed consent (e.g., by transparently recruiting participants through a crowdsourcing platform and requiring a consent form to be signed, as in ref. [3]), or without it (e.g., by responding to users' public posts and observing their reactions without disclosing the experiment taking place, as in ref. [4]).

Ethical concerns related to field experiments have long been discussed in the social sciences, but recent adaptations to social media platforms renew these concerns and raise new questions. Academic psychologists most familiar with experiments in controlled laboratory-type settings may quickly find themselves faced with ethical dilemmas that current guidelines struggle to address. For example, a central motivation for conducting a social media field experiment is to observe users' natural behaviours, so researchers may argue they should be excused from obtaining informed consent or permitted to deceive users for the sake of ecological validity. Such trade-offs mean that even researchers who follow guidance and carefully evaluate the harms and benefits of their study face many open questions and may unwittingly engage in ethical practices that spark controversy. Indeed, previously implemented field experiments on social media have raised ethical concerns and faced public scrutiny (Box 1). Such instances highlight the need for clearer ethical guidelines for academic experiments run on social media platforms, even if those experiments ultimately aim to shed light on the overarching issue of platforms' power over their users.

## Box 1. | Examples of ethical issues in social media experiments

Researchers conducting social media experiments may act in good faith, receive approval from an ethics committee and engage in ethical deliberation, including harms-benefits assessments as encouraged by some journals. For instance, the authors of ref. [15] discuss whether the benefits of their findings outweigh potential harms. However, previous field experiments on social media illustrate the ongoing challenges in protecting user rights and ensuring transparency, underscoring the need for clearer ethical guidelines, even if those experiments in question ultimately aim to shed light on the overarching issue of platforms' power over their users.

*Lack of informed consent*

In some cases, experiments have been conducted without obtaining informed consent from participants, raising questions about adherence to ethical standards. Facebook's 2014 *Emotional Contagion Experiment* involved nearly 700,000 users whose News Feeds were manipulated to assess emotional contagion without their knowledge or consent, sparking significant debate about informed consent and emotional manipulation[2].

*Deception of users*

Certain studies have employed deceptive practices, which, while methodologically justified in some cases, may risk eroding public trust. For example, one study on information spread in social networks created automated "bot" accounts to simulate human behaviour, avoiding Twitter's automated detection mechanisms. Though the hashtags promoted were prosocial (e.g., #getyourflushot), the deception involved raises ethical concerns about transparency and user trust[16].

*Interventions that promote unequal outcomes*

Field experiments may create real-world disparities, raising questions about fairness. For instance, a study examining social media promotion's impact on job market outcomes in economics randomized the promotion of job market papers by prominent economists. While the study provided valuable insights, it also created unequal benefits for participants, as those in the treatment group gained more views and job offers[15].

*Unintended consequences at scale*

Even the most well-intentioned experiments can have unforeseen consequences. For experiments that involve intervening on massive, real-world social networks, the unintended consequences can quickly propagate through the network and have far-reaching, undocumented

effects. A study set out to test the effects of publicly debunking posts containing links to false political news on Twitter[4]. After posting replies to 2000 users, it was observed that the intervention caused the users to spread more low-quality, partisan, and toxic content. This raises the question of whether social media field experiments should integrate phased-rollouts or early-stopping rules, since good intentions are not a reliable safeguard against unforeseen negative outcomes propagating through a social network.

Despite numerous calls to update ethical guidelines and the presence of ethical review processes[5–7], researchers are often left to make ad hoc design decisions on their own.

Here, we argue that revising ethical practices for social media experiments can benefit from a participatory, collective approach, specifically addressing issues that current ethical frameworks fail to capture in this rapidly changing landscape. Drawing inspiration from developments in data governance and artificial intelligence (AI), we advocate for democratising the creation of ethical guidelines and standards by taking the needs and concerns of researchers, platforms, and users into account.

## The problems in current ethical practices
Typically, psychological research within an academic setting that involves human participants is evaluated by an ethics committee, also known as institutional review boards (IRB)—unlike in industry. However, these traditional evaluators of ethics have struggled to keep pace with the rapid innovations in software and experimental designs developed for social media field experiments, leaving ethical boundaries blurred and dilemmas unresolved.

The general principles enforced by IRBs (e.g., minimisation of harm) straightforwardly apply to traditional laboratory experiments, but this is not the case for experiments conducted on active social media platforms. In social media contexts, issues like deception, consent, and real-time intervention are especially difficult to navigate. While some issues have been addressed (e.g., IRBs rightly do not view user acceptance of a platform's terms of service as informed consent to participate in an experiment), many other difficulties remain. For example, consider an experiment on the efficacy of an intervention to reduce toxic language on social media that uses automated accounts posing as humans to expose users to certain content. Should this deception be declared as a potential harm? Or, should deception via automated accounts be accepted as a regular occurrence online? And to what lengths should researchers go to ensure their online interventions do not exacerbate or introduce new vulnerabilities?

The current model suffers from a misalignment of interests. Researchers' interests in generating valuable data often conflict with the interests of social media users, who prioritise informed consent and autonomy, and the platforms, which are motivated by business goals and data monetisation. Where a researcher might argue that potential harms imposed by their experiment are commonplace on social media and outweighed by the potential impact of their research findings, the users involved in that same study may disagree and place a greater value on informed consent than potential research impact. Moreover, if the researcher collaborates with the involved platform, which has an interest in producing findings that portray the platform in a positive light, a conflict of interest can arise and open a door to industry capture of academic research if this is not explicitly declared. A further discussion of this can be found in the recent critique surrounding a high-profile *Science* paper[8], which counted former Meta employees as co-authors and suggested Facebook's algorithm plays little role in driving political polarisation—whilst failing to declare that the

platform implemented 63 changes to the algorithm during the experiment that were specifically geared towards reducing harmful content.

To ensure academic psychologists conduct ethical social media experiments, an approach that augments IRB processes is needed to incorporate the voices of all the stakeholders involved while remaining flexible enough to keep pace with evolving methodologies.

## How participatory approaches can help
Defined broadly as democratic engagement through deliberative practices, participatory governance emphasises citizen involvement in decision-making processes[9]. In a political context, this approach traditionally includes diverse methods such as public inquiries, citizen juries, and local councils, which seek to redistribute power and enhance democratic legitimacy. Parallel efforts in the realm of research include citizen science, broadly defined as public participation in scientific research, especially data collection[10].

At their core, the promise of participatory governance approaches and methods like citizen science lies in their ability to elicit collective intelligence and shared responsibility. For example, as recently applied to the field of AI[11,12], it advocates for incorporating broader public perspectives into the design and deployment of AI systems, ensuring alignment with societal values, and enhancing transparency and accountability. For instance, participatory design methodologies in AI involve end-users directly in the development process, bridging the gap between institutions and impacted communities through collective action. Success stories include establishing community data sovereignty protocols and participatory dataset documentation, such as Māori Data Sovereignty Protocols in New Zealand, which enabled the Māori community to take control of annotated audio data of their language, thereby preventing corporate entities getting hold of the dataset and using it to build commercial software products, such as speech-to-text technology, for their own profit[11].

In the case of social media research, a first aim may be for researchers to work with users to co-create standardised platform-appropriate and user-friendly methods to obtain informed consent, such as pop-up notifications or interactive tutorials, which improve transparency. Additionally, ethics committees may convene ethical 'sandboxing' whereby smaller groups of users participate in pilot versions of an experimental intervention (e.g., algorithmic nudges) and provide live feedback on whether the intervention feels intrusive, unfair, or disproportionately affects some users. It's important to stress that much of what makes social media so valuable for research (e.g., the ability to quickly change an intervention in real-time) can also be used to inform ethical practices.

By actively engaging users as co-producers in research, psychologists can address key concerns over consent, transparency, and potential harm in social media experiments, and ensure that research serves the public interest. Alongside designing consent processes collaboratively, this could involve conducting surveys or focus groups to identify user concerns about study design and ethical risks, and define early stopping rules in case unintended effects begin to manifest. For example, in a study on content moderation strategies, researchers could use focus groups to identify emotional distress from flagged content; based on this feedback, they might set an early stopping rule to pause the study if a significant number of participants report significant distress during initial testing. These practices help researchers embed inclusivity, equity, and empowerment into their methodologies[11]. This can arguably be especially instructive for studies involving topics or participants that belong to minorities or marginalised communities, which require additional considerations of how ethics, power, and risk to researchers themselves interact[13].

It must be recognised that any experiment conducted on a mainstream social media platform inherently operates in an ethically compromised landscape marked by a stark power imbalance between the platform and the researchers and users. In fact, examples of ethical concern in academic experiments (Box 1) often reflect this power imbalance (e.g., seemingly trivial interventions can have cascading effects through a social network unbeknownst to users). But this power imbalance should not discourage researchers and users from encouraging governments to require social media platforms to provide technical support if needed for participatory approaches to ethics to succeed. More fundamentally, governments themselves should not be intimidated by the demands of social media companies but rather continue to pursue measures that mitigate the risks of these platforms, especially the spread of online misinformation.

## Implementing a participatory approach

At first glance, a participatory approach to research ethics may appear idealistic. Researchers might be reluctant to adopt it because it seems overly time-consuming or resource intensive. Given the heterogeneity of social media users, it is also plausible that some types of users would be unwilling or unable to participate, even if there was an opportunity. However, decades of research in political science on citizen deliberation show that individuals, when provided with appropriate contexts and support, can effectively engage in complex decision-making processes. These studies suggest that institutional design, careful facilitation, and structured formats can mitigate challenges like resource demands and participant bias while ensuring fair outcomes[14]. So what might a more participatory approach actually look like for social media research?

There are several ways participatory approaches can help revise ethical practices (see Box 2). In this new paradigm, the process would involve multiple layers of participation, collaboration, and ongoing engagement between researchers, ethics committees, and social media users. Researchers, research institutions, and government bodies (e.g., NIH) would be responsible for setting up and overseeing the participatory process. For example, government mandates and institutional backing would ensure that the process is easily integrated into social media research workflows and academic training, ensuring early career researchers are encouraged to take up the cause. Online deliberation tools like Polis (https://pol.is/home) and AI-supported platforms could facilitate wide-reaching co-design workshops, where researchers and social media users collaborate to develop and continually update experimental protocols. In general, these co-design workshops would serve to open a dialogue between researchers and users—users can communicate what they view as requirements for being respected, and researchers can communicate what is technically feasible to implement whilst preserving ecological validity.

---

### Box 2. | Recommendations for improving ethical practices in online experimentation

Addressing ethical challenges in online experimentation requires fostering genuine collaboration while managing competing interests among stakeholders like researchers, platforms, and the public. Co-production mechanisms must explicitly counterbalance the influence of powerful actors (e.g., tech platforms) through transparent safeguards and independent oversight. To achieve inclusivity and relevance, recruiting diverse participants and tailoring panels to specific research topics are essential. Below, we provide some recommendations to meet these challenges and strengthen ethical practices.

*Co-design workshops with users*
- **Purpose**: Facilitate collaboration between researchers, IRBs, and social media users to co-create experimental protocols.
- **Process**: Develop and update risk assessment frameworks, clarify consent procedures, and integrate user perspectives on privacy and data use.
- **Safeguards**: Appoint external moderators to ensure open participation and prevent dominance by specific stakeholders

*Citizen juries and community advisory boards*
- **Citizen juries:** Randomly select social media users to deliberate on specific research proposals, ensuring demographic and geographic diversity. Include stipends and logistical support to encourage broad participation. Judgements may also be crowdsourced to inform the use of interventions.
- **Community advisory boards:** Engage advocacy groups, marginalized communities, and topic-specific experts (e.g., mental health advocates for psychological studies).
- **Safeguards:** Incorporate independent facilitators and external observers to mediate discussions and prevent undue influence from powerful stakeholders, including tech platforms. Set clear guidelines for decision-making and require public documentation of deliberations to enhance accountability.

*Digital ethics panels and oversight mechanisms*
- **Structure:** Create independent panels with balanced representation from academia, civil society, platforms, and the public. Panels should include ethicists and experts relevant to the research topic.
- **Safeguards:** Require transparent selection processes, public reporting of decisions, and external audits to mitigate the risk of "gaming" outcomes. Enforce conflict-of-interest disclosures and rotate membership to avoid entrenched biases.
- **Accountability:** Establish complaint mechanisms for participants and ensure panels have the authority to impose corrective actions for unethical practices.

*Tailored panels and research-specific mechanisms*
- **Flexibility:** Adapt panel composition based on the study's focus. For example:
  - Misinformation studies: Include journalists, fact-checkers, and media scholars.
  - Health-related experiments: Incorporate medical ethicists, patient advocacy groups, and healthcare professionals.
- **Recruitment:** Use targeted outreach to ensure panels reflect diverse perspectives, including underrepresented and vulnerable populations.

*Dynamic and user-driven engagement*
- **Feedback mechanisms:** Deploy live dashboards and periodic surveys to collect participant input throughout the research cycle.
- **Transparency:** Publish accessible summaries of ongoing research to enable public scrutiny and foster trust.
- **Adaptability:** Regularly update experimental designs based on real-time feedback to address emerging concerns and risks effectively.

---

Citizen juries composed of a diverse cross-section of social media users could review and deliberate on proposed experiments. These juries would serve as critical decision-making bodies, ensuring that all research proposals respect user autonomy and minimise harm. To maintain the balance of power, these deliberations would involve independent facilitators and expert

advisors from ethics committees (such as IRBs) who would ensure impartiality and prevent dominance by any one group. Drawing from models like patient advocacy groups in clinical research, which help shape study designs and recruit participants in rare diseases research, these user-focused bodies would aim to maintain up-to-date ethical guidance on public attitudes and could even be used to help recruit participants for research studies.

The outputs from the participatory initiatives we propose would in turn serve as inputs to ethics committees and IRBs. This would mean the latter retain their role per current legislation, but can turn to co-design workshops and citizen juries for up-to-date guidance on public attitudes. Whilst researchers would not be forced to engage in participatory initiatives, doing so would grant them insights that they could include in an IRB application, increasing their chances of receiving ethical approval and avoiding public backlash. Overall, the participatory approach to ethics we sketch out would ensure that researchers have the chance to engage more in ethical reflection and ethics committees and IRBs are themselves not simply gatekeepers, but active facilitators and participants in the process of ethical review.

In contexts where ethics committees or IRBs are absent, participatory initiatives could also still provide meaningful ethical insights. However, participatory initiatives alone cannot guarantee ethical rigour or enforce accountability in the absence of formal oversight mechanisms. Without the structured guidance and final approval of an ethics committee or IRB, there is a risk that such initiatives may lack consistency, impartiality, or the authority to prevent unethical practices. For these reasons, we argue that participatory initiatives should complement, not replace, formal ethics review processes. Given our proposals are targeted at academic researchers, establishing robust ethics committees or equivalent bodies remains a necessary foundational step.

Moving forward, as our recommendations do not immediately call for substantial regulatory change, their success depends on fostering a culture of continuous dialogue and reflection. Alongside our specific proposals, researchers and ethics committees should facilitate ongoing engagement with participants to assess the evolving ethical landscape and incorporate feedback into study designs. This iterative process ensures that ethical considerations remain at the forefront and that research practices adapt to changing societal expectations and concerns. Additionally, further promoting transparency in research practices through open communication and accessible dissemination of findings empowers participants and the public to understand and critique research outcomes.

Emphasising education and training on ethical conduct for researchers and participants alike further strengthens ethical practices, ensuring all stakeholders are equipped to navigate ethical dilemmas effectively. Finally, fostering interdisciplinary collaboration among psychologists, ethicists, technologists, and users promotes holistic approaches to ethical decision-making, integrating diverse perspectives and expertise to address complex ethical challenges responsibly.

Naturally, there are still several outstanding practical questions not covered here which require in-depth thought in order for participatory methods to be effective, such as funding considerations and time constraints. Regarding the former, introducing new taxes on social media platforms or increasing existing taxes, like the UK's Digital Services Tax, may be one route to consider. As for the latter, we suggest that our proposals are introduced gradually and follow a two-pronged approach, with processes integral to the review of active studies (i.e., the meetings of citizen juries and ethics committees) following a regular schedule, similar to data access committees in medical fields, for instance, which tend to process studies on a monthly basis. Meanwhile, those aimed at improving best practices (i.e., co-design workshops), could operate in a more agile manner

in response to new platform changes and continuously feed their findings into IRB meetings. By embracing these recommendations, psychologists can spearhead ethical standards in social media research, fostering trust, accountability, and societal benefit.

## Conclusion

Participatory approaches offer a promising solution to the ethical complexities of social media experiments by actively involving users as collaborators and embracing democratic deliberation. By integrating diverse perspectives, researchers can ensure that ethical guidelines reflect societal values, enhancing transparency, inclusivity, and accountability. However, much work remains to be done in testing and refining these methods. Practical challenges, such as engaging diverse stakeholders and addressing power imbalances, must be explored through ongoing iteration. As in other fields, the success of participatory ethics in social media research will depend on continuous adaptation and testing, ultimately fostering a research environment that benefits participants, researchers, and platforms alike while ensuring fair representation of all interests.

**Vincent J. Straub** [1] ✉ **& Jason W. Burton** [2,3] ✉
[1]Leverhulme Centre for Demographic Science, Nuffield Department of Population Health, University of Oxford, Oxford, United Kingdom.
[2]Department of Digitalization, Copenhagen Business School, Frederiksberg, Denmark. [3]Center for Adaptive Rationality, Max Planck Institute for Human Development Berlin, Berlin, Germany.
✉e-mail: vincent.straub@ndph.ox.ac.uk; jb.digi@cbs.dk

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

## Acknowledgements
This research was assisted by an SSRC/Summer Institutes in Computational Social Science (SICCS) Research Grant. The funder had no role in the decision to publish or prepare this manuscript. V.J.S. was funded by UK Research and Innovation (UKRI) under the UK government's Horizon Europe funding guarantee [EP/X027694/1] and the Leverhulme Trust Large Centre Grant LCDS (RC-2018-003). Both authors would also like to thank the reviewers and editor for their thoughtful suggestions, which greatly improved the manuscript.

## Author contributions
V.J.S. and J.W.B. conceptualised and drafted the manuscript together.

## Competing interests
The authors declare no competing interests.
