## [Peer Review file · Communications Psychology]

Participatory approaches should be used to address the ethics of social media experiments

Corresponding Author: Mr Vincent Straub

Version 0:

Decision Letter:

Dear Mr Straub,

Thank you for your patience during the peer-review process. Your manuscript titled "Collectively addressing the ethics of social media experiments" has now been seen by 3 reviewers, and I include their comments at the end of this message.

The reviewers are in principle enthusiastic about your work. However, they also mention a number of concerns. We are very interested in the possibility of publishing your manuscript in Communications Psychology, but would like to consider your response to these concerns in the form of a revised manuscript before we make a decision on publication.

To aid you with that task, I have included a marked-up version of your manuscript.

In sum, we invite you to revise your manuscript taking into account all reviewer and editor comments.

EDITORIAL POLICIES AND FORMATTING

You will find a complete list of formatting requirements following this link: <https://www.nature.com/documents/commsj-style-formatting-checklist-review-perspective.pdf>

Please use the checklist to prepare your manuscript for resubmission.

* **TRANSPARENT PEER REVIEW:** Communications Psychology uses a transparent peer review system. This means that we publish the editorial decision letters including Reviewers' comments to the authors and the author rebuttal letters online as a supplementary peer review file. We publish these records for all accepted manuscripts. However, on author request, confidential information and data can be removed from the published reviewer reports and rebuttal letters prior to publication. If your manuscript has been previously reviewed at another journal, those Reviewers' comments would not form part of the published peer review file.

If you have any questions about any of our policies or formatting, please don't hesitate to contact me.

Please use the following link to submit your revised manuscript and a point-by-point response to the referees' comments (which should be in a separate document to any cover letter):

Link Redacted

We hope to receive your revised paper within 12 weeks; please let us know if you aren't able to submit it within this time so that we can discuss how best to proceed. If we don't hear from you, and the revision process takes significantly longer, we may close your file.

We understand that due to the current global situation, the time required for revision may be longer than usual. We would appreciate it if you could keep us informed about an estimated timescale for resubmission, to facilitate our planning. Of course, if you are unable to estimate, we are happy to accommodate necessary extensions nevertheless.

Please do not hesitate to contact me if you have any questions or would like to discuss these revisions further. We look forward to seeing the revised manuscript and thank you for the opportunity to review your work.

Best regards,

Jennifer Bellingtier, PhD
Senior Editor
Communications Psychology

REVIEWERS' EXPERTISE:

Reviewer #1 social media studies
Reviewer #2 social media studies
Reviewer #3 social media studies

REVIEWERS' COMMENTS:

Reviewer #1 (Remarks to the Author):

Review of MS 24-0325

by Straub & Burton

Reviewer: Stephan Lewandowsky

Summary and Overall Recommendation

The article addresses the ethical issues that arise out of conducting field experiments on social media. As the authors correctly note, those methodologies are comparatively new and existing ethical guidelines may be insufficient to meet those challenges. The authors propose a new, participatory approach to dealing with those ethical issues. Participatory governance had shown some promise in adjacent fields such as AI.

The paper tackles an important issue and offers an interesting perspective on possible future developments, although at present the discussion appears lopsided to me because it largely ignores the current power structure of the online landscape. I think the impact of the paper can be considerably enhanced through revision but in principle I would be pleased to see this moving forward towards publication.

Major points

>> The authors do not consider the importance of the public interest in research findings. For example, when discussing the "misalignment" problem (p. 1, L 62-69), the authors mention the impact of research findings without exploring this further. Experiments that highlight the role of algorithms in shaping people's information diet are of critical public interest, and this should be acknowledged because it is perhaps the most important aspect of the ethical trade-offs facing researchers.

>> In general, the authors skirt the issue of corporate and state power. The only mention of power is in the context of researchers vis-à-vis marginalized communities (p. 3, L 109), which is indeed a relevant point, but it is dwarfed by the monumental power imbalance that exists between platforms and users. This power imbalance is ethically deeply problematic. It is impossible to fully discuss research ethics without recognizing that most researchers are operating in a landscape that is already ethically compromised by the platforms. Arguably, some of the examples in Box 1 are helpful in illustrating that power imbalance, and the ethical issues arising from that research may well be outweighed by the insights it provides on the greater over-arching ethical issue of platform power.

>> The authors provide a text box (Box 2) that is outlining their proposal. The box seems underdeveloped. Much more could (and should) be said here. For starters, the authors should connect more to the extensive literature on deliberative assemblies in political science, e.g. DOI 10.1126/science.aaw2694, which underscores the importance of expert input and moderation to deliberative fora (see also <https://sks.to/techdem> Chapter 7 for a review). The need for independent moderation and expert input should probably feature in Box 2.

There is also no recognition of the tension between the interests of various stakeholders (e.g., platforms vs. public) and how those might be resolved during co-production. What mechanisms are required to prevent the most powerful actors (viz. the tech platforms) from dominating ethics panels? How can "gaming" of the outcome be prevented? How would people be selected to serve on those panels?

The devil here is very much in the detail and for this paper to have the impact it deserves, some of those details must be, at the very least, spelled out even if their resolution must await future discussions.

Detailed comments

[page#]:[line#]

1:19 The study by Kozyreva et al. should probably be cited here as well because it provides an in-depth look at people's attitudes towards data use in three countries. (DOI: 10.1057/s41599-021-00787-w).

2:82 "existent"? Not sure what the intended meaning is.

Reviewer #2 (Remarks to the Author):

This is an interesting comment piece on an important growing topic. The comment is well-written. I do have several feedbacks that the authors need to address before publications.

- While conducting field experiments on social media is relatively new to psychological science, similar ethical challenges have existed for a variety field experiments across other fields e.g., in terms of whether to obtain informed, potential unintended consequences, deception etc. It is important the authors point out this caveat.

- It is also important to mention the trade-off between violating ecological validity versus decisions to whether inform the participants, use deception etc. That is, if the researchers let the participants know they are part of an experiment participants might behave differently and the results might not be ecologically valid anymore. Yet, the researchers need to figure out the potential ethical consequences for such decisions.

- The participatory approach suggested by the authors has a good potential to partially address the ethical challenges. However, they might be resource intensive and refrain researchers from adopting such approach. Additionally, given the heterogeneity of participants across experiments, it might not be easy to involve similar users in the participatory design or the group of users who will be the participants in the research will not be accessible through other means. For example, extreme users who are responsible for sharing problematic content online might not be willing to participate in the co-designing an experiment.

- The authors' description of study 21 in Box 1 doesn't seem accurate. That study is not about exposing users to misinformation or conspiracy theories, yet about debunking false content by providing truthful information from fact-checking website. Thus, the goal of that study was to improve users behavior/belief. Therefore, concluding that that experiment was ethically concerning would imply that any field experiment on social media (or any experiment anywhere) is ethically concerning, because some kind of unintended effect might occur.

Reviewer #3 (Remarks to the Author):

This piece presents the problem of a lack of clear ethical guidelines around certain types of social media field experiments, and proposed a participatory approach to creating or updating such guidelines.

First, I think that the first paragraph lacks some clarity and specificity that would be helpful for many readers. I think that largest issue is that it is unclear whether social media field experiment explicitly means an experiment conducted without consent. And then it may be confusing, because it's non-obvious how a researcher would conduct such an experiment without cooperation of the platform. In any case, I think that it would be immensely helpful to have a specific, concrete example that is explained right at the front here. Or perhaps multiple examples - the Facebook emotional contagion study is more well known and is an example of such an experiment conducted with/by the platform, but I think that other such field experiments (e.g. the Twitter hashtags experiment) are less obvious. (Until I read Box 1 I was actually struggling to think of one myself!)

Also re: clarity, I wasn't sure if "researchers" and "academic psychologists" were being used interchangeably here or if researchers was inclusive of the researchers who work for social media platforms. I also am not sure about the meaning of the last sentence; is "while" being used to indicate contrast?

The statement "academic studies require approval from an institutional review board" is incorrect; it would be more accurate to say that human subjects research studies require approval from an IRB, though also it should be noted that IRB is a U.S.-centric term. (IRB vs ERB vs the lack of such review systems in some contexts entirely should probably be explicitly stated in the more detailed discussion of IRBs in the next section.)

I also think that an issue with the IRB section is that I think there are some questions posed that we have answers to. For example, there were very specific reasons that the Cornell IRB did not review the FB emotional contagion study, and we know what they were. And under the Common Rule, anything that constitutes "intervention" requires informed consent. (And I've never heard of any IRB considering TOS as relevant at all to consent.) If these actually are open questions that we know about, it would be helpful to cite some prior work. I'm not as familiar with work on IRB attitudes with field experiments, but there is a fair amount about IRBs and public data.

It also seems a bit disconnected that this piece focuses so much on IRBs without explicitly stating why they are not sufficient to deal with this issue (which I think is an argument that could be made) and also how they would fit into the new paradigm that is proposed here.

I actually think that the idea of a participatory approach here is a good one, and the concept is fairly well laid out. A shortcoming however is that it's unclear what this looks like in practice. Who is running this? Who is enforcing it? And what ethical guidelines are being updated, exactly? Is this a suggestion for changes to IRBs, or something in the profession of psychology?

In summary, I love the general idea proposed here of a user-involved participatory approach to ethical guidelines for social media experimentation in psychology. However, the presentation of the problem lacks clarity, as do the practical components of the proposed solution.

Version 1:

Decision Letter:

Dear Mr Straub,

Your manuscript titled "Collectively addressing the ethics of social media experiments" has now been seen by our reviewers, whose comments appear below. In light of their advice I am delighted to say that we are happy, in principle, to publish a suitably revised version in Communications Psychology.

We therefore invite you to revise your paper one last time to address the remaining concerns of our reviewers and a list of editorial requests. At the same time we ask that you edit your manuscript to comply with our format requirements and to maximise the accessibility and therefore the impact of your work.

EDITORIAL REQUESTS:

Please review our specific editorial comments and requests regarding your manuscript in the attached edited manuscript and "Editorial Requests Table". Please outline your response to each request in the right hand column. Please upload the completed table with your manuscript files as a Related Manuscript file. Please also see the attached edited manuscript file.

SUBMISSION INFORMATION:

OPEN ACCESS:

At acceptance, you will be provided with instructions for completing the open access licence agreement on behalf of all

authors. This grants us the necessary permissions to publish your paper. Additionally, you will be asked to declare that all required third party permissions have been obtained, and to provide billing information in order to pay the article-processing charge (APC).

Link Redacted

Best regards,

Jennifer Bellingtier

Jennifer Bellingtier, PhD
Senior Editor
Communications Psychology

REVIEWERS' EXPERTISE:

Reviewer #1 social media studies
Reviewer #2 social media studies
Reviewer #3 social media studies

REVIEWERS' COMMENTS:

Reviewer #1 (Remarks to the Author):

Review of MS 24-0325A

by Straub & Burton

Reviewer: Stephan Lewandowsky

Summary and Overall Recommendation

The article addresses the ethical issues that arise out of conducting field experiments on social media. As the authors correctly note, those methodologies are comparatively new and existing ethical guidelines may be insufficient to meet those challenges. The authors propose a new, participatory approach to dealing with those ethical issues. Participatory governance had shown some promise in adjacent fields such as AI.

I was positively inclined towards this paper at the first round, and the revision has improved considerably. I am happy to recommend publication although I believe another round of revision (without further review) is required to deal with the following issues.

Major points

>> One issue with the participatory approach that is left untouched is (a) who is funding this process and (b) how it can be conducted in a timely manner. Researchers cannot wait 6 months to constitute some deliberative body and work through ethics with that body for a single study. There are lots of devils in the details here, and this needs to be acknowledged.

Detailed comments

[line#] (in marked-up version)

34-35 "... excused from obtaining informed consent and permitted to deceive users for the sake of ecological validity." Those are two separate issues: omitting informed consent need not involve deception. It is possible to observe behaviour in vivo without deception but also without explicit consent.

117 This would be a good place to mention the recent controversy surrounding the Meta-funded research, where it turned out that Facebook had modified the newsfeed algorithm in a control condition during a field experiment, thus compromising

interpretability of the data. This is an instance where declaration of funding is insufficient because the research was compromised by industry action (it is contested whether or not the researchers were aware of Facebook's action). See <https://www.science.org/content/article/study-found-facebook-algorithm-didnt-promote-political-polarization-critics-doubt> and links therein for this controversy.

142-143 Please explain the success story in 1-2 sentences.

158-160 This is still a very abstract level of explanation. Are there any concrete examples that could be reported here? It still requires considerable cognitive work (and imagination) by the reader to figure out what is meant here.

213 This is the first time platforms are mentioned in the preceding paragraphs outlining the co-production idea. It is unclear why platforms should be involved in this process at all (except to ensure technical feasibility perhaps, which has nothing to do with ethics). The ethical issues affect users and must be resolved by researchers and relevant institutions (i.e., IRB and so on).

226 Ethics instead of ethical?

Reviewer #2 (Remarks to the Author):

The authors have sufficiently accommodated my comments.

Reviewer #3 (Remarks to the Author):

The description of ethical review is still not completely accurate. Even in the U.S., not every psychology researcher has to submit an application to a government-mandated body of reviewers. For example, if a psychology researcher worked for Meta or for Microsoft, this would not be the case. (And this of course was part of the issue with the Facebook emotional contagion study and how the experiment itself was not reviewed by the Cornell IRB because only Facebook researchers were engaged in the actual data collection.) Also there are some things here that I think could use references or additional context, e.g. the relationship between TOS and informed consent. I also think that the role of IRBs is still a bit confusing going into the next section - I wasn't sure if "a new approach is needed" was an explicit statement that IRBs should be replaced, or added to, or something else?

I also still like the general participatory proposal, but still don't understand how this would work in practice. How would researchers work with users to create method for informed consent, if those methods require design features of the platform, like pop-ups or interactive tutorials? This implies to me that the researchers have to be working with the platform, which is an entirely different issue. I think the idea of involving users in understanding user concerns is great (and there's been a lot of prior work on this related to researcher use of public data), and even designing consent practices, but in order for this to make sense these have to be consent practices that researchers can actually implement. Or there needs to be some reflection on the challenge of that implementation if it would require action on the part of the platforms. Later there is a mention of government bodies, but if the intention here is for all of this to require new platform regulation, I think that is a really big deal and shouldn't just be glossed over. I do think it's possible to make this suggestion (and as the paper notes, it is idealistic) if the intention here is to be provocative, but I think that this component needs to be more front and center, that either (a) the interventions need to be things that researchers can implement, such as best practices or using external tools; or (b) this is just as much about regulating platforms as about creating research norms.

I also think that the clarity with respect to the new role of ethics committees/IRBs is useful, but it is important to note that this would require substantial regulatory change as well. The purview and function of IRBs is very explicitly laid out in the Common Rule (45 CFR 46). Also, how would this whole process work in the absence of ethics committees - for example, in countries without bodies such as IRBs, or psychology research taking place by the platforms themselves (like the emotional contagion study)?

The additional clarity in this revision did make the proposal much easier to understand, and again I do think that the participatory idea is a good one, but I find the details of the proposal to be confusing in some places and unrealistic or lacking in necessary context in others.

Response to Review of MS 24-0325 by Straub & Burton

Point-to-point response to reviewers in boxes below

REVIEWERS' COMMENTS:

Reviewer #1 (Remarks to the Author):

Reviewer: Stephan Lewandowsky

Summary and Overall Recommendation

The article addresses the ethical issues that arise out of conducting field experiments on social media. As the authors correctly note, those methodologies are comparatively new and existing ethical guidelines may be insufficient to meet those challenges. The authors propose a new, participatory approach to dealing with those ethical issues. Participatory governance has shown some promise in adjacent fields like AI. The paper tackles an important issue and offers an interesting perspective on possible future developments. Currently, the discussion seems to me because it largely ignores the current power structure of the online landscape. I think the impact of the paper can be considerably enhanced through revision but in principle I would be pleased to see this moving forward towards publication.

We appreciate the reviewer's thoughtful comments and are grateful for their recognition of the importance of addressing the ethical challenges of field experiments on social media and exploring participatory governance approaches. We also agree that the current power structures of the online landscape represent a critical issue that underpins many of these ethical challenges.

While our article focuses on the ethical considerations and practical application of participatory governance in social media research, we acknowledge that addressing the broader power dynamics of major social media platforms is essential for creating a more equitable online research environment. As the reviewer notes, these power structures have significant implications for the governance of social media experiments. However, we believe this topic goes beyond the scope and primary aim of our (short) commentary.

That said, we wholeheartedly agree that this is an important area for further scholarly exploration, particularly by legal and policy experts. Breaking up or more effectively regulating dominant social media companies is a needed and much-discussed legal argument in this regard. Such measures could complement the participatory ethics

framework we propose, ensuring that any reforms are not overshadowed or undermined by the overwhelming influence of powerful platforms.

We hope this clarification highlights the intended scope of our article while acknowledging the importance of broader systemic considerations. Thank you for bringing this critical issue to our attention.

Major points

>> The authors do not consider the importance of the public interest in research findings. For example, when discussing the “misalignment” problem (p. 1, L 62-69), the authors mention the impact of research findings without exploring this further. Experiments that highlight the role of algorithms in shaping people’s information diet are of critical public interest, and this should be acknowledged because it is perhaps the most important aspect of the ethical trade-offs facing researchers.

Thank you for raising this point. We fully agree that social media field experiments can benefit the public interest and that this potential should factor into researchers' consideration of ethical tradeoffs. The participatory approach we advocate for is designed to facilitate the navigation of this tradeoff by allowing researchers to directly engage with the public to better understand and serve their interests. Instead of researchers paternalistically deciding what qualifies as being in the public interest, initiatives like community advisory boards (Box 2) offer a new way to involve the public in the decision-making process around ethical approval. Intuitively, it would seem more compelling for a study to justify bypassing standard practices, like informed consent or debriefing, in the name of public interest if that justification has been vetted by a board of platform users, rather than a committee of academics in “the ivory tower.” In doing so, there is also an opportunity to establish a sense of shared responsibility in cases where an experimental intervention elicits unforeseen, unintended negative effects.

In response to this point, we have revised the text to more explicitly link the participatory approach for which we advocate to the public interest:

“By actively engaging users as co-producers in research, psychologists can address key concerns over consent, transparency, and potential harm in social media experiments, and ensure that research serves the public interest” (p. 3).

>> In general, the authors skirt the issue of corporate and state power. The only mention of power is in the context of researchers vis-à-vis marginalized communities (p. 3, L 109), which is indeed a relevant point, but it is dwarfed by the monumental power imbalance that exists between platforms and users. This power imbalance is ethically

deeply problematic. It is impossible to fully discuss research ethics without recognizing that most researchers are operating in a landscape that is already ethically compromised by the platforms. Arguably, some of the examples in Box 1 are helpful in illustrating that power imbalance, and the ethical issues arising from that research may well be outweighed by the insights it provides on the greater overarching ethical issue of platform power.

This is an excellent issue to highlight. The academic research community faces a dilemma: How can research be done to expose the platform-user power imbalance on social media without engaging with and contributing to that imbalance? Indeed, the examples in Box 1 illustrate that power imbalance — e.g., it's trivial for a platform to conduct experiments on users without their consent and trigger unintended consequences at scale. However, it seems like a slippery slope to allow the ends to justify the means of academic research. Even if academic researchers mean well and aim to expose ethically concerning practices of social media platforms, those researchers should not be waived of their ethical responsibilities, especially since it is often difficult to prove any tangible benefit to the manipulated participants.

Nevertheless, the reviewer is absolutely correct to raise this point, and it should be mentioned in our manuscript. We have thus revised the text as follows:

“However, it must also be recognized that any experiment conducted on a mainstream social media platform inherently operates in an ethically compromised landscape marked by a stark power imbalance between the platform and the users. In fact, examples of ethical concern in academic experiments (Box 1) often reflect this power imbalance (e.g., seemingly trivial interventions can have cascading effects through a social network unbeknownst to users)” (p. 3).

“Researchers conducting social media experiments may act in good faith, receive approval from an ethics committee and engage in ethical deliberation, including harms-benefits assessments as encouraged by certain journals. For instance, the authors of ref. 14 discuss whether the benefits of their findings outweigh potential harms. However, previous field experiments on social media illustrate the ongoing challenges in protecting user rights and ensuring transparency, underscoring the need for clearer ethical guidelines, even if those experiments in question ultimately aim to shed light on the overarching issue of platforms’ power over their users” (Box 1).

>> The authors provide a text box (Box 2) that is outlining their proposal. The box seems underdeveloped. Much more could (and should) be said here. For starters, the authors should connect more to the extensive literature on deliberative assemblies in political science, e.g. DOI 10.1126/science.aaw2694, which underscores the importance of expert input and moderation to deliberative fora (see also <https://sks.to/techdem>

Chapter 7 for a review). The need for independent moderation and expert input should probably feature in Box 2.

There is also no recognition of the tension between the interests of various stakeholders (e.g., platforms vs. public) and how those might be resolved during co-production. What mechanisms are required to prevent the most powerful actors (viz. the tech platforms) from dominating ethics panels? How can “gaming” of the outcome be prevented? How would people be selected to serve on those panels?

The devil here is very much in the details and for this paper to have the impact it deserves, some of those details must be, at the very least, spelled out even if their resolution must await future discussions.

We thank the reviewer for pointing out that the box of recommendations can benefit from further development and for providing links to additional literature on political deliberation. We have sought to now acknowledge all of the points put forth and updated Box 2 so that it is a lot more comprehensive and reads as follows:

“Box 2. Recommendations for improving ethical practices in online experimentation

Addressing ethical challenges in online experimentation requires fostering genuine collaboration while managing competing interests among stakeholders like researchers, platforms, and the public. Co-production mechanisms must explicitly counterbalance the influence of powerful actors (e.g., tech platforms) through transparent safeguards and independent oversight. To achieve inclusivity and relevance, recruiting diverse participants and tailoring panels to specific research topics are essential. Below, we provide some recommendations to meet these challenges and strengthen ethical practices.

Co-design workshops with users

- **Purpose:** Facilitate collaboration between researchers, IRBs, and social media users to co-create experimental protocols.
- **Process:** Develop and update risk assessment frameworks, clarify consent procedures, and integrate user perspectives on privacy and data use.
- **Safeguards:** Appoint external moderators to ensure open participation and prevent dominance by specific stakeholders

Citizen juries and community advisory boards

- **Citizen juries:** Randomly select social media users to deliberate on specific research proposals, ensuring demographic and geographic diversity. Include stipends and logistical support to encourage broad participation.

- **Community advisory boards:** Engage advocacy groups, marginalized communities, and topic-specific experts (e.g., mental health advocates for psychological studies).
- **Safeguards:** Incorporate independent facilitators and external observers to mediate discussions and prevent undue influence from powerful stakeholders, including tech platforms. Set clear guidelines for decision-making and require public documentation of deliberations to enhance accountability.

Digital ethics panels and oversight mechanisms

- **Structure:** Create independent panels with balanced representation from academia, civil society, platforms, and the public. Panels should include ethicists and experts relevant to the research topic.
- **Safeguards:** Require transparent selection processes, public reporting of decisions, and external audits to mitigate the risk of "gaming" outcomes. Enforce conflict-of-interest disclosures and rotate membership to avoid entrenched biases.
- **Accountability:** Establish complaint mechanisms for participants and ensure panels have the authority to impose corrective actions for unethical practices.

Tailored panels and research-specific mechanisms

- **Flexibility:** Adapt panel composition based on the study's focus. For example:
 - Misinformation studies: Include journalists, fact-checkers, and media scholars.
 - Health-related experiments: Incorporate medical ethicists, patient advocacy groups, and healthcare professionals.
- **Recruitment:** Use targeted outreach to ensure panels reflect diverse perspectives, including underrepresented and vulnerable populations.

Dynamic and user-driven engagement

- **Feedback mechanisms:** Deploy live dashboards and periodic surveys to collect participant input throughout the research cycle.
- **Transparency:** Publish accessible summaries of ongoing research to enable public scrutiny and foster trust.
- **Adaptability:** Regularly update experimental designs based on real-time feedback to address emerging concerns and risks effectively."

1:19 The study by Kozyreva et al. should probably be cited here as well because it provides an in-depth look at people's attitudes towards data use in three countries. (DOI: 10.1057/s41599-021-00787-w).

This citation has now been incorporated. We thank the reviewer for the suggestion.

2:82 “existent”? Not sure what the intended meaning is.

This has been replaced with the word ‘established’ to clarify the meaning that we are referring to efforts already in place elsewhere. We thank the reviewer for the suggestion.

Reviewer #2 (Remarks to the Author):

This is an interesting commentary on an important topic. The comment is well-written. I do have several feedbacks that the authors need to address before publication.

- While conducting field experiments on social media is relatively new to psychological science, similar ethical challenges have existed for a variety of field experiments across other fields e.g., in terms of whether to obtain information, potential unintended consequences, deception etc. It is important the authors point out this caveat.

We agree that ethical concerns around field experiments have been considered by social scientists for a long time, and it’s important for us to point that out. However, the widespread use of *social media* field experiment seems like quite a recent trend among academic researchers, hence the recent posting of several papers offering methodological guidelines, such as Mosleh, Pennycook, & Rand (2021, <https://doi.org/10.1177/09637214211054761>), Aridor et al. (2024, <http://dx.doi.org/10.2139/ssrn.4991773>), and Piccardi et al. (2024, <https://doi.org/10.48550/arXiv.2406.19571>).

In response to this point, we have revised the opening paragraph of our manuscript as follows:

“Ethical concerns related to field experiments have long been discussed in the social sciences, but recent adaptations of the methodology to social media platforms renew these concerns and raise new questions. Academic psychologists most familiar with experiments in controlled laboratory-type settings may quickly find themselves faced with ethical dilemmas that current guidelines struggle to address” (p. 1).

- It is also important to mention the trade-off between violating ecological validity versus decisions to inform the participants, use deception etc. That is, if the researchers let the

participants know they are part of an experiment, participants might behave differently and the results might not be ecologically valid anymore. Still, researchers must find out the potential ethical consequences for such decisions.

Thank you for raising this point. The tradeoff between informed consent/deception and ecological validity is particularly salient when conducting social media field experiments.

In response to this point, we have revised the text in the opening paragraph as follows:

“Academic psychologists most familiar with experiments in controlled laboratory-type settings may quickly find themselves faced with ethical dilemmas and tradeoffs that current guidelines struggle to address. For example, informed consent is easily obtained and non-negotiable when participants are invited to a physical lab or to a purpose-built online experiment implemented with Qualtrics, jsPsych, or similar web application. Yet, a central motivation for conducting a social media field experiment is to observe users’ natural behaviours, so researchers may argue they should be excused from obtaining informed consent and permitted to deceive users for the sake of ecological validity. Such tradeoffs mean that even researchers who follow guidance and carefully evaluate harms and benefits of their study face many open questions and may unwittingly engage in ethical practices that spark controversy (see Box 1)” (p. 1).

- The participatory approach suggested by the authors has a good potential to partially address the ethical challenges. However, they might be resource intensive and refrain researchers from adopting such an approach. Additionally, given the heterogeneity of participants across experiments, it might not be easy to involve similar users in the participatory design or the group of users who will be the participants in the research will not be accessible through other means. For example, extreme users who are responsible for sharing problematic content online might not be willing to participate in the co-designing of an experiment.

We thank the reviewer for their comment and note that they raise important considerations which we previously did not fully acknowledge. We have now sought to integrate these considerations into our revised manuscript by adding an additional paragraph to the subsection ‘Recommendations for a participatory ethics’, which mentions each of the concerns above, and expanded Box 2, which discusses each and states how they may be addressed.

- The authors' description of study 21 in Box 1 doesn't seem accurate. That study is not about exposing users to misinformation or conspiracy theories, but about debunking false content by providing truthful information from fact-checking websites. Thus, the goal of that study was to improve users behavior/belief. Therefore, concluding that that experiment was ethically concerning would imply that any field experiment on social media (or any experiment anywhere) is ethically concerning, because some kind of unintended effect might occur.

Thank you for raising this point, which alludes to the well-known ethical dilemma between deontology and consequentialism: Should the good intentions of a researcher take precedent over unintended negative outcomes of the researcher's experiment? We agree with the reviewer that this dilemma affects just about any experiment anywhere. However, we do not think it is clear that a strictly deontological perspective should be adhered to in the context of social media field experiments.

In traditional lab settings, unintended consequences of well-intentioned experiments are typically limited to a relatively small number of participants and regular ethical safeguards, like informed consent and debriefing, are readily implemented. With social media field experiments, thousands of participants can be directly affected by an unintended consequence (and potentially many thousands more indirectly through the social network), and safeguards like informed consent and debriefing are often not easy to implement. These differences suggest that the ethical perspective that "works" for traditional lab settings might not be directly transferable to social media field experiments.

Consider the particular paper mentioned, Mosleh et al. (2021, <https://doi.org/10.1145/3411764.3445642>). We admire the researchers who conducted the study and we do not question their intentions. Nevertheless, the study involved 2,000 Twitter users without their informed consent, provided no debriefing, and ultimately caused the users to spread more low-quality, partisan, and toxic content. The scale and reach of such unintended consequences is concerning, regardless of the researchers' reasonable rationale for prioritizing ecological validity and users' anonymity.

Over the past decades, small sample sizes have been widely criticized in psychology due to concerns around statistical power. Nowadays, social media field experiments that intervene in massive, real-world social networks seem to raise different concerns: Can a sample size be too large? Does a particular intervention *need* to be tested with a social media field experiment, or could it just as well be tested in an isolated, "sandbox" environment? Perhaps it is time for psychologists to borrow techniques like phased roll-outs and early stopping rules, which have long been part of clinical trials and large-scale A/B testing. In fact, in the EU, GDPR dictates that the collection of personal data must be limited to what is necessary (i.e., the principle of data minimization) (Greene, Shmueli, Ray, & Fell, 2019,

<https://doi.org/10.1089/big.2018.0176>). Our commentary does not claim to have the answers, but rather intends to start a conversation and involve all relevant stakeholders.

In response to this point, we have revised the text in Box 1 to more accurately reflect the potential ethical concern around Mosleh et al. (2021, <https://doi.org/10.1145/3411764.3445642>):

“Unintended consequences at scale:

Even the most well-intentioned experiments can have unforeseen consequences. For experiments that involve intervening on massive, real-world social networks, the unintended consequences can quickly propagate through the network and have far-reaching, undocumented effects. A study set out to test the effects of publicly debunking posts containing links to false political news on Twitter [4]. After posting replies to 2,000 users, it was observed that the intervention caused the users to spread more low-quality, partisan, and toxic content. This raises the question of whether social media field experiments should integrate phased-rollouts or early-stopping rules, since good intentions are not a reliable safeguard against unforeseen negative outcomes propagating through a social network” (Box 1).

Reviewer #3 (Remarks to the Author):

This piece presents the problem of a lack of clear ethical guidelines around certain types of social media field experiments, and proposes a participatory approach to creating or updating such guidelines.

First, I think that the first paragraph lacks some clarity and specificity that would be helpful for many readers. I think the biggest problem is that it is unclear whether social media field experiment explicitly means an experiment conducted without consent. And then it may be confusing, because it's non-obvious how a researcher would conduct such an experiment without cooperation of the platform. In any case, I think that it would be immensely helpful to have a specific, concrete example that is explained right at the front here. Or perhaps multiple examples - the Facebook emotional contagion study is more well known and is an example of such an experiment conducted with/by the platform, but I think that other such field experiments (e.g. the Twitter hashtags experiment) are less obvious. (Until I read Box 1 I was actually struggling to think of one myself!)

Thank you for raising this point. The social media field experiments we refer to in our manuscript are not strictly those without informed consent. While it intuitively seems easier to conduct a social media field experiment without informed consent if the researchers have explicit collaboration with the platform (as in the Facebook emotional contagion study), it is also possible to conduct such experiments without platform collaboration. For example, Mønsted et al. (2017, <https://doi.org/10.1371/journal.pone.0184148>) created automated “bot” account to directly interact with thousands of users without seeking informed consent or providing a debriefing (no ethical justification is provided in the paper); Mosleh et al. (2021, <https://doi.org/10.1145/3411764.3445642>) directly replied to public posts on Twitter to observe the original posters’ reaction without seeking informed consent or providing a debriefing (they justify this as necessary for maintaining ecological validity and participants’ anonymity).

In response to this point, we have revised the text to clarify the scope of experiments we consider and to highlight some exemplary references (while aiming to adhere to the word count and reference limitations):

“The experiments can be run with platform cooperation [e.g., with direct industry-academia collaboration, as in Kramer, Guillory, & Hancock (2014) and Guess et al (2023)] or independently, without explicit collaboration agreements with the platform on which they are run [e.g., by having consenting users download web browser extensions that scrape posts from their feed, as in Milli et al. (2023) and Levy (2021)]. The experiments can be run with participants’ informed consent [e.g., by transparently recruiting participants through a crowdsourcing platform and requiring a consent form be signed, as in Milli et al. (2023)] or without it (e.g., by straightforwardly responding to users’ public posts and observing their reactions without disclosing the experiment taking place, as in Mosleh et al. (2021)]” (p. 1).

Also re: clarity, I wasn't sure if "researchers" and "academic psychologists" were being used interchangeably here or if "researchers" was inclusive of the researchers who work for social media platforms. I also am not sure about the meaning of the last sentence; is "while" being used to indicate contrast?

We thank the reviewer for pointing out this ambiguous phrasing and have sort to rectify it accordingly:

“Academic researchers make subjective, ethical judgments when designing experiments on human participants. This is particularly evident in the decisions psychologists must make when designing and implementing field experiments on social media” (p. 1)

The statement "academic studies require approval from an institutional review board" is incorrect. It would be more accurate to say that human subjects research studies require approval from an IRB, though also it should be noted that IRB is a U.S.-centric term. (IRB vs ERB vs the lack of such review systems in some contexts entirely should probably be explicitly stated in the more detailed discussion of IRBs in the next section.)

Thank you for pointing out the inaccuracy and regional specificity of our original statement regarding IRB approval for academic studies. We agree with the reviewer's observation that it would be more precise to specify that human subjects research requires approval from an IRB (or equivalent ethics review body) and that IRB terminology is primarily used in U.S. contexts. In light of this we have changed the phrasing and wording in our revised version (see section The problems in current ethical practices):

"In most countries, if a psychology researcher wants to run a study involving human participants, they must first submit an application detailing their experimental design to an independent, government-mandated body of reviewers that aims to ensure the research adheres to ethical standards. In practice, a local committee typically administers this process at the researcher's own institution, typically known as institutional review boards (IRB) in the US and ethics committees elsewhere, with each differing slightly in their scope depending on the country (we use both terms interchangeably throughout for fluency)..." (p. 1)

I also think that an issue with the IRB section is that I think there are some questions posed that we have answers to. For example, there were very specific reasons that the Cornell IRB did not review the FB emotional contagion study, and we know what they were. And under the Common Rule, anything that constitutes "intervention" requires informed consent. (And I've never heard of any IRB considering TOS as relevant at all to consent.) If these actually are open questions that we know about, it would be helpful to cite some prior work. I'm not as familiar with work on IRB attitudes with field experiments, but there is a fair amount about IRBs and public data.

Thank you for highlighting the need to refine our discussion on IRBs and their role in social media research. We appreciate your observation regarding the specific reasons behind the Cornell IRB's decision not to review the Facebook emotional contagion study and the requirements under the Common Rule for interventions to include informed consent. In response, we have updated the text (see below) to acknowledge that some of the questions we originally posed, such as the ones you reference, do have answers in existing literature. We now emphasize that while certain aspects of IRB decisions and regulations are understood, other critical questions, particularly

those specific to field experiments on social media, remain unresolved. To enhance the discussion, we have also added citations to prior work on IRBs and public data to contextualize our points. We believe these changes will clarify the scope of our inquiry and improve the precision of our arguments. Thank you for your insightful suggestions.

“While some issues have been resolved (e.g., IRBs rightly do not view user acceptance of a platform’s terms of service as informed consent to participate in an experiment), many other difficulties remain. For example, if an experiment on the efficacy of an intervention to reduce toxic language on social media uses automated accounts posing as humans to expose users to certain content, should this deception be declared as a potential harm? Or, should deception via automated accounts be accepted as a regular occurrence online? And to what lengths should researchers go to ensure their online interventions do not exacerbate or introduce new vulnerabilities?” (p. 2)

It also seems a bit disconnected that this piece focuses so much on IRBs without explicitly stating why they are not sufficient to deal with this issue (which I think is an argument that could be made) and also how they would fit into the new paradigm that is proposed here.

Thank you for your valuable feedback. We agree that the piece could benefit from a clearer explanation of why IRBs, as they currently function, may not be sufficient to address the ethical challenges of social media research. In response, we have now explicitly acknowledged this limitation in the revised manuscript. We also provide additional citations to support this argument and have clarified the role that IRBs and ethics committees may play in the new, participatory paradigm we propose. Specifically, in the section Implementing a Participatory Approach, we explain how ethics committees could evolve to serve as facilitators and moderators within a more inclusive and deliberative framework. This revision aims to better integrate the discussion of IRBs with the new ethical model and to more directly address their potential role in the future of social media research ethics.

I actually think that the idea of a participatory approach here is a good one, and the concept is fairly well laid out. A shortcoming however is that it's unclear what this looks like in practice. Who is running this? Who is enforcing it? And what ethical guidelines are being updated? Is this a suggestion for changes to IRBs, or something in the profession of psychology?

Thank you for your thoughtful comment. We agree that providing more clarity on the

practical implementation of the participatory approach is essential. In response, we have revised the manuscript to explicitly outline the roles of various stakeholders in this new paradigm, including researchers, ethics committees, and social media users. We clarify how ethics committees, such as IRBs, would play a key role in moderating the process and ensuring fairness. Additionally, we provide more details on how ongoing engagement and feedback loops would help enforce ethical standards and update guidelines to reflect the evolving landscape of social media research. These updates are now incorporated throughout in the section on "Implementing a Participatory Approach." We hope this addresses your concerns and provides a clearer picture of how this approach could be enacted.

In summary, I love the general idea proposed here of a user-involved participatory approach to ethical guidelines for social media experimentation in psychology. However, the presentation of the problem lacks clarity, as do the practical components of the proposed solution.

Thank you for your positive feedback on the concept of a user-involved participatory approach. We appreciate your insight regarding the clarity of the problem presentation and the practical components of the proposed solution. In response, we have revised the manuscript to provide a clearer and more structured explanation of the issues at hand. Additionally, we have enhanced the practical aspects of the proposed solution by offering more detailed examples of how the participatory approach could be implemented, including the roles of key stakeholders, the involvement of users, and how ethics committees and ongoing feedback mechanisms would operate (see updated Box 2). We hope these revisions address your concerns and reinforce the clarity of the argument.

REVIEWERS' COMMENTS:

Reviewer #1 (Remarks to the Author):

Review of MS 24-0325A

by Straub & Burton

Reviewer: Stephan Lewandowsky

Summary and Overall Recommendation

The article addresses the ethical issues that arise out of conducting field experiments on social media. As the authors correctly note, those methodologies are comparatively new and existing ethical guidelines may be insufficient to meet those challenges. The authors propose a new, participatory approach to dealing with those ethical issues. Participatory governance had shown some promise in adjacent fields such as AI.

I was positively inclined towards this paper at the first round, and the revision has improved considerably. I am happy to recommend publication although I believe another round of revision (without further review) is required to deal with the following issues.

Major points

>> One issue with the participatory approach that is left untouched is (a) who is funding this process and (b) how it can be conducted in a timely manner. Researchers cannot wait 6 months to constitute some deliberative body and work through ethics with that body for a single study. There are lots of devils in the details here, and this needs to be acknowledged.

We thank the reviewer for pointing out these missing points. Whilst we are unable to fully address them due to space constraints, we have added an additional paragraph that speaks to the point raised:

"Naturally, there are still several outstanding practical questions not covered here which require in-depth thought in order for participatory methods to be effective, such as funding considerations and time constraints. Regarding the former, introducing new taxes on social media platforms or increasing existing taxes, like the UK's Digital Services Tax, may be one route to consider. As for the latter, we suggest that our proposals are introduced gradually and follow a two-pronged approach, with processes integral to the review of active studies (i.e., the meetings of citizen juries and ethics committees) following a regular schedule, similar to data access committees in medical fields, for instance, which tend to process studies on a monthly basis, whilst those aimed at improving best practices (i.e., co-design workshops), occurring in a more agile manner in response to new platform changes and subsequently feeding their findings into future committee meetings. By embracing these recommendations, psychologists can spearhead ethical standards in social media research, fostering trust, accountability, and societal benefit."

Detailed comments

[line#] (in marked-up version)

34-35 "... excused from obtaining informed consent and permitted to deceive users for the sake of ecological validity." Those are two separate issues: omitting informed consent need not involve deception. It is possible to observe behaviour in vivo without deception but also without explicit consent.

This has been changed to acknowledged both are separate issues.

117 This would be a good place to mention the recent controversy surrounding the Meta-funded research, where it turned out that Facebook had modified the newsfeed algorithm in a control condition during a field experiment, thus compromising interpretability of the data. This is an instance where declaration of funding is insufficient because the research was compromised by industry action (it is contested whether or not the researchers were aware of Facebook's action). See <https://www.science.org/content/article/study-found-facebook-algorithm-didnt-promote-political-polarization-critics-doubt> and links therein for this controversy.

We thank the reviewer for drawing our attention to this recent controversy and have cited it in the relevant section, the edited sentence now reads as follows:

"Moreover, if the researcher collaborates with the involved platform, which has an interest in producing findings that portray the platform in a positive light, a conflict of interest can arise and open a door to industry capture of academic research if this is not explicitly declared. A further discussion of this can be found in the recent critique surrounding a high-profile Science paper [8], which counted former Meta employees as co-authors and suggested Facebook's algorithm plays little role in driving political polarization whilst failing to declare that the algorithm in question changed during the time the study was conducted."

142-143 Please explain the success story in 1-2 sentences.

We have added an additional sentence to contextualise the success story, which reads as follows:

"Success stories include establishing community data sovereignty protocols and participatory dataset documentation, such as Māori Data Sovereignty Protocols in New Zealand, which enabled Māori community to take control of recordings of annotated hours of audio data of their language, thereby preventing corporate entities getting hold of the dataset to build products [10]."

158-160 "Alongside designing consent processes collaboratively, this could involve conducting surveys or focus groups to identify user concerns about study design and ethical risks, and define early stopping rules in case unintended effects begin to manifest..." This is still a very abstract level of explanation. Are there any concrete examples that could be reported here? It still requires considerable cognitive work (and imagination) by the reader to figure out what is meant here.

We thank the reviewer for their comment. We agree that concrete examples are useful to make our point and have included one to make our point more clearly, which now reads as:

"Alongside designing consent processes collaboratively, this could involve conducting surveys or focus groups to identify user concerns about study design and ethical risks, and define early stopping rules in case unintended effects begin to manifest. For example, in a study on content moderation strategies, researchers could use focus groups to identify emotional distress from flagged content; based on this feedback, they might set an early stopping rule to pause the study if a significant number of participants report significant distress during initial testing.

213 This is the first time platforms are mentioned in the preceding paragraphs outlining the co-production idea. It is unclear why platforms should be involved in this process at all (except to ensure technical feasibility perhaps, which has nothing to do with ethics). The ethical issues affect users and must be resolved by researchers and relevant institutions (i.e., IRB and so on).

We thank the reviewer for pointing this out, which was an error on our part; we have now revised the text at various points to clarify that the role of social media platforms is to provide the technical support to implement the changes we propose and as a potential source of funding via new regulatory measures.

226 Ethics instead of ethical?

We thank the reviewer for catching this error, which has now been corrected.

Reviewer #3 (Remarks to the Author):

The description of ethical review is still not completely accurate. Even in the U.S., not every psychology researcher has to submit an application to a government-mandated body of reviewers. For example, if a psychology researcher worked for Meta or for Microsoft, this would not be the case. (And this of course was part of the issue with the Facebook emotional contagion study and how the experiment itself was not reviewed by the Cornell IRB because only Facebook researchers were engaged in the actual data collection.) Also there are some things here that I think could use references or additional context, e.g. the relationship between TOS and informed consent. I also think that the role of IRBs is still a bit confusing going into the next section - I wasn't sure if "a new approach is needed" was an explicit statement that IRBs should be replaced, or added to, or something else?

We thank the reviewer for pointing out that the description of ethical review is still not completely accurate, the wish for extra context between TOS and informed consent, and clarification around the last sentence of the first/second section. We have sought to address each of these in turn by providing additional sentences or clarifying phrases that we hope adequately address the reviewer's concerns. For example, the last sentence of the first section now reads:

"To ensure academic psychologists conduct ethical social media experiments, an approach that augments IRB processes is needed to incorporate the voices of all the stakeholders involved while remaining flexible enough to keep pace with evolving methodologies."

And the last sentence of the second section now reads:

"...But this power imbalance should not discourage researchers and users from encouraging governments to require social media platforms to provide technical support if needed for participatory approaches to ethics to succeed."

I also still like the general participatory proposal, but still don't understand how this would work in practice. How would researchers work with users to create method for informed consent, if those methods require design features of the platform, like pop-ups or interactive tutorials? This implies to me that the researchers have to be working with the platform, which is an entirely different issue. I think the idea of involving users in understanding user concerns is great (and there's been a lot of prior work on this related to researcher use of public data), and even designing consent practices, but in order for this to make sense these have to be consent practices that researchers can actually implement. Or there needs to be some reflection on the challenge of that implementation if it would require action on the part of the platforms. Later there is a mention of government bodies, but if the intention here is for all of this to require new platform regulation, I think that is a really big deal and shouldn't just be glossed over. I do think it's possible to make this suggestion (and as the paper notes, it is idealistic) if the intention here is to be provocative, but I think that this component needs to be more front and center, that either (a) the interventions need to be

things that researchers can implement, such as best practices or using external tools; or (b) this is just as much about regulating platforms as about creating research norms.

We thank the reviewer for pushing us to clarify how the participatory proposal would work in practice. We do not believe that working with the platforms is necessary. Indeed, several papers have already demonstrated that it is possible to conduct a social media field experiment and gain informed consent without platform collaboration (e.g., Milli et al., 2023; Levy, 2021; Piccardi et al., 2024). While we agree that it can be technically unfeasible for some design features to be implemented without platform collaboration (e.g., deploying pop-ups directly in the UI of a social media feed), this is precisely why our participatory approach is appealing. Current ethical practices do not require any dialogue between researchers and users. Yet, through initiatives like co-design workshops (Box 2), researchers can communicate what is and is not feasible directly to users, and both parties can debate the best way to navigate trade-offs between things like deception, ecological validity, and avoiding conflicts of interest by partnering with the platforms themselves. Put simply, our participatory proposal works by moving debates over ethical practices between researchers and users to the planning stage of an experiment, rather than having that debate *after* an experiment has been published, when it is too late for adjustments to be made. We have sought to clarify this point with the following revision in the manuscript:

“In general, these co-design workshops would serve to open a dialogue between researchers and users, so that users can communicate what they view as requirements for being respected, and researchers can communicate what is technically feasible to implement whilst preserving ecological validity.”

With regards to platform regulation, we do believe that that is an important aspect to consider when thinking about ethical practices and the future of social media experiments. However, we view this as a tangential topic that we cannot cover in depth while respecting the length restrictions for a commentary paper. We hope to connect these topics on a deeper level in future work.

References:

Levy, R. E. (2021). Social media, news consumption, and polarization: Evidence from a field experiment. *American economic review*, 111(3), 831-870.

Milli, S., Carroll, M., Wang, Y., Pandey, S., Zhao, S., & Dragan, A. D. (2023). Engagement, user satisfaction, and the amplification of divisive content on social media. *arXiv preprint arXiv:2305.16941*.

Piccardi, T., Saveski, M., Jia, C., Hancock, J. T., Tsai, J. L., & Bernstein, M. (2024). Social Media Algorithms Can Shape Affective Polarization via Exposure to Antidemocratic Attitudes and Partisan Animosity. *arXiv preprint arXiv:2411.14652*.

I also think that the clarity with respect to the new role of ethics committees/IRBs is useful, but it is important to note that this would require substantial regulatory change as well. The

purview and function of IRBs is very explicitly laid out in the Common Rule (45 CFR 46). Also, how would this whole process work in the absence of ethics committees - for example, in countries without bodies such as IRBs, or psychology research taking place by the platforms themselves (like the emotional contagion study)?

We thank the reviewer for raising these questions. In the short-term, we view the initiatives within our participatory proposal (see Box 2) as augmenting ethics committees/IRBs. For example, the outputs from a co-design workshop could serve as inputs to an IRB's decisions about whether to grant ethical approval to applicants. In this way, substantial regulatory change is not immediately required. While regulatory change may be required in the longer term to prevent researchers from neglecting or evading up-to-date recommendations elicited by participatory initiatives (e.g., by "IRB shopping," Spellacy & May, 2021), we view this as an issue to address at a later stage. We have sought to clarify this point with the following revision in the manuscript:

"The outputs from the participatory initiatives we propose would in turn serve as inputs to ethics committees and IRBs. This would mean the latter retain their role per current legislation, but can turn to co-design workshops and citizen juries for up-to-date guidance on public attitudes. Whilst researchers would not be enforced to engage in participatory initiatives, doing so would grant them insights that they could include in an IRB application, increasing their chances of receiving ethical approval and avoiding public backlash. Overall, the participatory approach to ethics we sketch out would ensure that researchers have the chance to engage more in ethical reflection and ethics committees and IRBs are themselves not simply gatekeepers, but active facilitators and participants in the process of ethical review."

Where ethics committees/IRBs are absent, participatory initiatives could still hold value. For instance, it seems likely that the Facebook researchers planning the emotional contagion study would have benefited from first submitting their plans to a community advisory board. However, without oversight or final approval provided by an ethics committee/IRB, it could not be guaranteed that the participatory initiative was conducted properly. For this reason, we do not believe participatory initiatives would suffice on their own; ethics committees/IRBs would need to be established first, as pointed out in a new paragraph we have added:

"In contexts where ethics committees or IRBs are absent, participatory initiatives could still provide meaningful ethical insights. However, participatory initiatives alone cannot guarantee ethical rigor or enforce accountability in the absence of formal oversight mechanisms. Without the structured guidance and final approval of an ethics committee or IRB, there is a risk that such initiatives may lack consistency, impartiality, or the authority to prevent unethical practices. For these reasons, we argue that participatory initiatives should complement, not replace, formal ethics review processes. Establishing robust ethics committees or equivalent bodies remains a necessary foundational step."

We also stress that our proposal is targeted at academic researchers in our commentary.

Reference:

Spellecy, R., & May, T. (2012). More than cheating: Deception, IRB shopping, and the normative legitimacy of IRBs. *Journal of Law, Medicine & Ethics*, 40(4), 990-996.

The additional clarify in this revision did make the proposal much easier to understand, and again I do think that the participatory idea is a good one, but I find the details of the proposal to be confusing in some places and unrealistic or lacking in necessary context in others.

We thank the reviewer for their comments. We believe the revisions they have guided, such as those above, have clarified details of our proposal and strengthened the manuscript overall.